# FlashAttention-2: Faster Attention with Better Parallelism and Work Partitioning

**Tri Dao**[1][2]
[1]Department of Computer Science, Princeton University
[2]Department of Computer Science, Stanford University
`tridao@princeton.edu`

## Abstract

Scaling Transformers to longer sequence lengths has been a major problem in the last several years, promising to improve performance in language modeling and high-resolution image understanding, as well as to unlock new applications in code, audio, and video generation. The attention layer is the main bottleneck in scaling to longer sequences, as its runtime and memory increase quadratically in the sequence length. FlashAttention (Dao et al., 2022) exploits the asymmetric GPU memory hierarchy to bring significant memory saving (linear instead of quadratic) and runtime speedup (2-4× compared to optimized baselines), with no approximation. However, FlashAttention is still not nearly as fast as optimized matrix-multiply (GEMM) operations, reaching only 25-40% of the theoretical maximum FLOPs/s. We observe that the inefficiency is due to suboptimal work partitioning between different thread blocks and warps on the GPU, causing either low-occupancy or unnecessary shared memory reads/writes. We propose FlashAttention-2, with better work partitioning to address these issues. In particular, we (1) tweak the algorithm to reduce the number of non-matmul FLOPs (2) parallelize the attention computation, even for a single head, across different thread blocks to increase occupancy, and (3) within each thread block, distribute the work between warps to reduce communication through shared memory. These yield around 2× speedup compared to FlashAttention, reaching 50-73% of the theoretical maximum FLOPs/s on A100 and getting close to the efficiency of GEMM operations. We empirically validate that when used end-to-end to train GPT-style models, FlashAttention-2 reaches training speed of up to 225 TFLOPs/s per A100 GPU (72% model FLOPs utilization).

## 1 Introduction

Scaling up the context length of Transformers (Vaswani et al., 2017) is a challenge, since the attention layer at their heart has runtime and memory requirements quadratic in the input sequence length. Ideally, we would like to go beyond the standard 2k sequence length limit to train models to understand books, high resolution images, and long-form videos. Just within the last year, there have been several language models with much longer context than before: GPT-4 (OpenAI, 2023) with context length 32k, MosaicML's MPT with context length 65k, and Anthropic's Claude with context length 100k. Emerging use cases such as long document querying and story writing have demonstrated a need for models with such long context.

To reduce the computational requirement of attention on such long context, there have been numerous methods proposed to approximate attention (Kitaev et al., 2020; Roy et al., 2021; Wang et al., 2020; Katharopoulos et al., 2020; Choromanski et al., 2020; Beltagy et al., 2020; Zaheer et al., 2020; Chen et al., 2021). Though these methods have seen some use cases, as far as we know, most large-scale training runs still use standard attention. Motivated by this, Dao et al. (2022) proposed to reorder the attention computation and leverages classical techniques (tiling, recomputation) to significantly speed it up and reduce memory usage from quadratic to linear in sequence length. This yields 2-4× wall-clock time speedup over optimized baselines, up to 10-20× memory saving, with no approximation, and as a result FlashAttention has seen wide adoption in large-scale training and inference of Transformers.

However, context length increases even more, FlashAttention is still not nearly as efficient as other primitives such as matrix-multiply (GEMM). In particular, while FlashAttention is already 2-4× faster than a standard attention implementation, the forward pass only reaches 30-50% of the

theoretical maximum FLOPs/s of the device (Fig. 6), while the backward pass is even more challenging, reaching only 25-35% of maximum throughput on A100 GPU (Fig. 7). In contrast, optimized GEMM can reach up to 80-90% of the theoretical maximum device throughput. Through careful profiling, we observe that FLASHATTENTION still has suboptimal work partitioning between different thread blocks and warps on the GPU, causing either low-occupancy or unnecessary shared memory reads/writes.

Building on FLASHATTENTION, we propose FLASHATTENTION-2 with better parallelism and work partitioning to address these challenges.

1. In Section 3.1, we tweak the algorithms to reduce the number of non-matmul FLOPs while not changing the output. While the non-matmul FLOPs only account for a small fraction of the total FLOPs, they take longer to perform as GPUs have specialized units for matrix multiply, and as a result the matmul throughput can be up to 16× higher than non-matmul throughput. It is thus important to reduce non-matmul FLOPs and spend as much time as possible doing matmul FLOPs.

2. We propose to parallelize both the forward pass and backward pass along the sequence length dimension, in addition to the batch and number of heads dimension. This increases occupancy (utilization of GPU resources) in the case where the sequences are long (and hence batch size is often small).

3. Even within one block of attention computation, we partition the work between different warps of a thread block to reduce communication and shared memory reads/writes.

In Section 4, we empirically validate that FLASHATTENTION-2 yields significant speedup compared to even FLASHATTENTION. Benchmarks on different settings (with or without causal mask, different head dimensions) show that FLASHATTENTION-2 achieves around 2× speedup over FLASHATTENTION, reaching up to 73% of the theoretical max throughput in the forward pass, and up to 63% of the theoretical max throughput in the backward pass. During LLM inference, FLASHATTENTION-2's kernel is up to 7× faster than the attention kernel from FasterTransformer. When used end-to-end to train GPT-style models, we reach training speed of up to 225 TFLOPs/s per A100 GPU.

## 2   BACKGROUND

We provide some background on the performance characteristics and execution model of GPUs. We also describe the standard implementation of attention, as well as FLASHATTENTION.

### 2.1   HARDWARE CHARACTERISTICS

**GPU performance characteristics.** The GPU consists of compute elements (e.g., floating point arithmetic units) and a memory hierarchy. Most modern GPUs contain specialized units to accelerate matrix multiply in low-precision (e.g., Tensor Cores on Nvidia GPUs for FP16/BF16 matrix multiply). The memory hierarchy comprise of high bandwidth memory (HBM), and on-chip SRAM (aka shared memory). As an example, the A100 GPU has 40-80GB of high bandwidth memory (HBM) with bandwidth 1.5-2.0TB/s and 192KB of on-chip SRAM per each of 108 streaming multiprocessors with bandwidth estimated around 19TB/s (Jia et al., 2018; Jia and Van Sandt, 2021). As the L2 cache is not directly controllable by the programmer, we focus on the HBM and SRAM for the purpose of this discussion.

**Execution Model.** GPUs have a massive number of threads to execute an operation (called a kernel). Threads are organized into thread blocks, which are scheduled to run on streaming multiprocessors (SMs). Within each thread blocks, threads are grouped into warps (a group of 32 threads). Threads within a warp can communicate by fast shuffle instructions or cooperate to perform matrix multiply. Warps within a thread block can communicate by reading from / writing to shared memory. Each kernel loads inputs from HBM to registers and SRAM, computes, then writes outputs to HBM.

### 2.2   STANDARD ATTENTION IMPLEMENTATION

Given input sequences $\mathbf{Q}, \mathbf{K}, \mathbf{V} \in \mathbb{R}^{N \times d}$ where $N$ is the sequence length and $d$ is the head dimension, we want to compute the attention output $\mathbf{O} \in \mathbb{R}^{N \times d}$:

$$\mathbf{S} = \mathbf{Q}\mathbf{K}^\top \in \mathbb{R}^{N \times N}, \quad \mathbf{P} = \mathrm{softmax}(\mathbf{S}) \in \mathbb{R}^{N \times N}, \quad \mathbf{O} = \mathbf{P}\mathbf{V} \in \mathbb{R}^{N \times d},$$

where softmax is applied row-wise.[1] For multi-head attention (MHA), this same computation is performed in parallel across many heads, and parallel over the batch dimension (number of input sequences in a batch).

---

[1] For clarity of exposition, we omit the scaling of $\mathbf{Q}\mathbf{K}^\top$ (typically by $1/d$), and optionally elementwise masking on $\mathbf{S}$ and/or dropout applied to $\mathbf{P}$

The backward pass of attention proceeds as follows. Let $\mathbf{dO} \in \mathbb{R}^{N \times d}$ be the gradient of $\mathbf{O}$ with respect to some loss function. Then by the chain rule (aka backpropagation):

$$\mathbf{dV} = \mathbf{P}^\top \mathbf{dO} \in \mathbb{R}^{N \times d} \qquad \mathbf{dP} = \mathbf{dO}\mathbf{V}^\top \in \mathbb{R}^{N \times N}$$

$$\mathbf{dS} = \mathrm{dsoftmax}(\mathbf{dP}) \in \mathbb{R}^{N \times N} \qquad \mathbf{dQ} = \mathbf{dS}\mathbf{K} \in \mathbb{R}^{N \times d} \qquad \mathbf{dK} = \mathbf{dS}^\top \mathbf{Q} \in \mathbb{R}^{N \times d},$$

where dsoftmax is the gradient (backward pass) of softmax applied row-wise. One can work out that if $p = \mathrm{softmax}(s)$ for some vector $s$ and $p$, then with output gradient $dp$, the input gradient $ds = (\mathrm{diag}(p) - pp^\top)dp$.

Standard attention implementations materialize the matrices $\mathbf{S}$ and $\mathbf{P}$ to HBM, which takes $O(N^2)$ memory. Often $N \gg d$ (typically $N$ is on the order of 1k–8k and $d$ is around 64–128). The standard attention implementation (1) calls the matrix multiply (GEMM) subroutine to multiply $\mathbf{S} = \mathbf{QK}^\top$, writes the result to HBM, then (2) loads $\mathbf{S}$ from HBM to compute softmax and write the result $\mathbf{P}$ to HBM, and finally (3) calls GEMM to get $\mathbf{O} = \mathbf{PV}$. As most of the operations are bounded by memory bandwidth, the large number of memory accesses translates to slow wall-clock time. Moreover, the required memory is $O(N^2)$ due to having to materialize $\mathbf{S}$ and $\mathbf{P}$. Moreover, one has to save $\mathbf{P} \in \mathbb{R}^{N \times N}$ for the backward pass to compute the gradients.

## 2.3 FLASHATTENTION

To speed up attention on hardware accelerators such as GPU, (Dao et al., 2022) proposes an algorithm to reduce the memory reads/writes while maintaining the same output (without approximation).

### 2.3.1 FORWARD PASS

FLASHATTENTION applies the classical technique of tiling to reduce memory IOs, by (1) loading blocks of inputs from HBM to SRAM, (2) computing attention with respect to that block, and then (3) updating the output without writing the large intermediate matrices $\mathbf{S}$ and $\mathbf{P}$ to HBM. As the softmax couples entire rows or blocks of row, online softmax (Milakov and Gimelshein, 2018; Rabe and Staats, 2021) can split the attention computation into blocks, and rescale the output of each block to finally get the right result (with no approximation). By significantly reducing the amount of memory reads/writes, FLASHATTENTION yields 2-4× wall-clock speedup over optimized baseline attention implementations.

We describe the online softmax technique (Milakov and Gimelshein, 2018) and how it is used in attention (Rabe and Staats, 2021). For simplicity, consider just one row block of the attention matrix $\mathbf{S}$, of the form $\begin{bmatrix} \mathbf{S}^{(1)} & \mathbf{S}^{(2)} \end{bmatrix}$ for some matrices $\mathbf{S}^{(1)}, \mathbf{S}^{(2)} \in \mathbb{R}^{B_r \times B_c}$, where $B_r$ and $B_c$ are the row and column block sizes. We want to compute softmax of this row block and multiply with the value, of the form $\begin{bmatrix} \mathbf{V}^{(1)} \\ \mathbf{V}^{(2)} \end{bmatrix}$ for some matrices $\mathbf{V}^{(1)}, \mathbf{V}^{(2)} \in \mathbb{R}^{B_c \times d}$. Standard softmax would compute:

$$m = \max(\mathrm{rowmax}(\mathbf{S}^{(1)}), \mathrm{rowmax}(\mathbf{S}^{(2)})) \in \mathbb{R}^{B_r} \qquad \ell = \mathrm{rowsum}(e^{\mathbf{S}^{(1)}-m}) + \mathrm{rowsum}(e^{\mathbf{S}^{(2)}-m}) \in \mathbb{R}^{B_r}$$

$$\mathbf{P} = \begin{bmatrix} \mathbf{P}^{(1)} & \mathbf{P}^{(2)} \end{bmatrix} = \mathrm{diag}(\ell)^{-1} \begin{bmatrix} e^{\mathbf{S}^{(1)}-m} & e^{\mathbf{S}^{(2)}-m} \end{bmatrix} \in \mathbb{R}^{B_r \times 2B_c}$$

$$\mathbf{O} = \begin{bmatrix} \mathbf{P}^{(1)} & \mathbf{P}^{(2)} \end{bmatrix} \begin{bmatrix} \mathbf{V}^{(1)} \\ \mathbf{V}^{(2)} \end{bmatrix} = \mathrm{diag}(\ell)^{-1} e^{\mathbf{S}^{(1)}-m} \mathbf{V}^{(1)} + e^{\mathbf{S}^{(2)}-m} \mathbf{V}^{(2)} \in \mathbb{R}^{B_r \times d}.$$

Online softmax instead computes "local" softmax with respect to each block and rescale to get the right output at the end:

$$m^{(1)} = \mathrm{rowmax}(\mathbf{S}^{(1)}) \in \mathbb{R}^{B_r} \qquad \ell^{(1)} = \mathrm{rowsum}(e^{\mathbf{S}^{(1)}-m^{(1)}}) \in \mathbb{R}^{B_r}$$

$$\tilde{\mathbf{P}}^{(1)} = \mathrm{diag}(\ell^{(1)})^{-1} e^{\mathbf{S}^{(1)}-m^{(1)}} \in \mathbb{R}^{B_r \times B_c} \qquad \mathbf{O}^{(1)} = \tilde{\mathbf{P}}^{(1)} \mathbf{V}^{(1)} = \mathrm{diag}(\ell^{(1)})^{-1} e^{\mathbf{S}^{(1)}-m^{(1)}} \mathbf{V}^{(1)} \in \mathbb{R}^{B_r \times d}$$

$$m^{(2)} = \max(m^{(1)}, \mathrm{rowmax}(\mathbf{S}^{(2)})) = m$$

$$\ell^{(2)} = e^{m^{(1)}-m^{(2)}} \ell^{(1)} + \mathrm{rowsum}(e^{\mathbf{S}^{(2)}-m^{(2)}}) = \mathrm{rowsum}(e^{\mathbf{S}^{(1)}-m}) + \mathrm{rowsum}(e^{\mathbf{S}^{(2)}-m}) = \ell$$

$$\tilde{\mathbf{P}}^{(2)} = \mathrm{diag}(\ell^{(2)})^{-1} e^{\mathbf{S}^{(2)}-m^{(2)}}$$

$$\mathbf{O}^{(2)} = \mathrm{diag}(\ell^{(1)}/\ell^{(2)}) \mathbf{O}^{(1)} + \tilde{\mathbf{P}}^{(2)} \mathbf{V}^{(2)} = \mathrm{diag}(\ell^{(2)})^{-1} e^{s^{(1)}-m} \mathbf{V}^{(1)} + \mathrm{diag}(\ell^{(2)})^{-1} e^{s^{(2)}-m} \mathbf{V}^{(2)} = \mathbf{O}.$$

We show how FLASHATTENTION uses online softmax to enable tiling (Fig. 1) to reduce memory reads/writes.

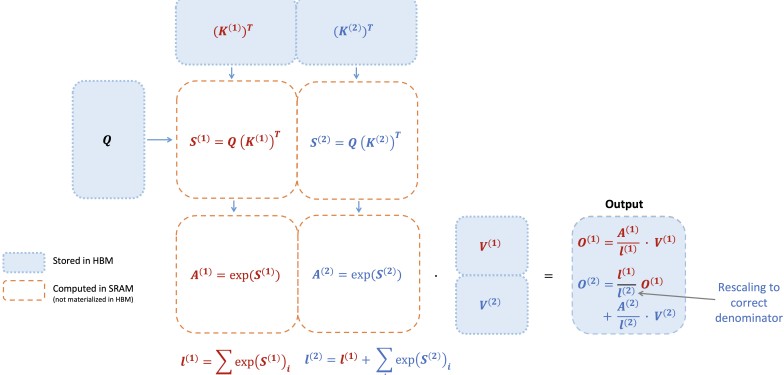

Figure 1: Diagram of how FLASHATTENTION forward pass is performed, when the key **K** is partitioned into two blocks and the value **V** is also partitioned into two blocks. By computing attention with respect to each block and rescaling the output, we get the right answer at the end, while avoiding expensive memory reads/writes of the intermediate matrices **S** and **P**. We simplify the diagram, omitting the step in softmax that subtracts each element by the row-wise max.

### 2.3.2   BACKWARD PASS

In the backward pass, by re-computing the values of the attention matrices **S** and **P** once blocks of inputs **Q**, **K**, **V** are already loaded to SRAM, FLASHATTENTION avoids having to store large intermediate values. By not having to save the large matrices **S** and **P** of size $N \times N$, FLASHATTENTION yields 10-20× memory saving depending on sequence length (memory required in linear in sequence length $N$ instead of quadratic). The backward pass also achieves 2-4× wall-clock speedup due to reduce memory reads/writes.

The backward pass applies tiling to the equations in Section 2.2. Though the backward pass is simpler than the forward pass conceptually (there is no softmax rescaling), the implementation is significantly more involved. This is because there are more values to be kept in SRAM to perform 5 matrix multiples in the backward pass, compared to just 2 matrix multiples in the forward pass.

## 3   FLASHATTENTION-2: ALGORITHM, PARALLELISM, AND WORK PARTITIONING

We describe the FLASHATTENTION-2 algorithm, which includes several tweaks to FLASHATTENTION to reduce the number of non-matmul FLOPs. We then describe how to parallelize the computation on different thread blocks to make full use the GPU resources. Finally we describe we partition the work between different warps within one thread block to reduce the amount of shared memory access. These improvements lead to 2-3× speedup as validated in Section 4.

### 3.1   ALGORITHM

We tweak the algorithm from FLASHATTENTION to reduce the number of non-matmul FLOPs. This is because modern GPUs have specialized compute units (e.g., Tensor Cores on Nvidia GPUs) that makes matmul much faster. As an example, the A100 GPU has a max theoretical throughput of 312 TFLOPs/s of FP16/BF16 matmul, but only 19.5 TFLOPs/s of non-matmul FP32. Another way to think about this is that each non-matmul FLOP is 16× more expensive than a matmul FLOP. To maintain high throughput (e.g., more than 50% of the maximum theoretical TFLOPs/s), we want to spend as much time on matmul FLOPs as possible.

#### 3.1.1   FORWARD PASS

We revisit the online softmax trick as shown in Section 2.3 and make two minor tweaks to reduce non-matmul FLOPs:

1. We do not have to rescale both terms of the output update by $\text{diag}(\ell^{(2)})^{-1}$:

$$\mathbf{O}^{(2)} = \text{diag}(\ell^{(1)}/\ell^{(2)})\mathbf{O}^{(1)} + \text{diag}(\ell^{(2)})^{-1}e^{\mathbf{S}^{(2)}-m^{(2)}}\mathbf{V}^{(2)}.$$

We can instead maintain an "un-scaled" version of $\mathbf{O}^{(2)}$ and keep around the statistics $\ell^{(2)}$:

$$\tilde{\mathbf{O}}^{(2)} = \mathrm{diag}(\ell^{(1)})^{-1}\mathbf{O}^{(1)} + e^{\mathbf{S}^{(2)}-m^{(2)}}\mathbf{V}^{(2)}.$$

Only at the every end of the loop do we scale the final $\tilde{\mathbf{O}}^{(\mathrm{last})}$ by $\mathrm{diag}(\ell^{(\mathrm{last})})^{-1}$ to get the right output.

2. We do not have to save both the max $m^{(j)}$ and the sum of exponentials $\ell^{(j)}$ for the backward pass. We only need to store the logsumexp $L^{(j)} = m^{(j)} + \log(\ell^{(j)})$.

In the simple case of 2 blocks in Section 2.3, the online softmax trick now becomes:

$$m^{(1)} = \mathrm{rowmax}(\mathbf{S}^{(1)}) \in \mathbb{R}^{B_r} \qquad \ell^{(1)} = \mathrm{rowsum}(e^{\mathbf{S}^{(1)}-m^{(1)}}) \in \mathbb{R}^{B_r}$$

$$\mathbf{O}^{\tilde{(1)}} = e^{\mathbf{S}^{(1)}-m^{(1)}}\mathbf{V}^{(1)} \in \mathbb{R}^{B_r \times d} \qquad m^{(2)} = \max(m^{(1)}, \mathrm{rowmax}(\mathbf{S}^{(2)})) = m$$

$$\ell^{(2)} = e^{m^{(1)}-m^{(2)}}\ell^{(1)} + \mathrm{rowsum}(e^{\mathbf{S}^{(2)}-m^{(2)}}) = \mathrm{rowsum}(e^{\mathbf{S}^{(1)}-m}) + \mathrm{rowsum}(e^{\mathbf{S}^{(2)}-m}) = \ell$$

$$\tilde{\mathbf{P}}^{(2)} = \mathrm{diag}(\ell^{(2)})^{-1}e^{\mathbf{S}^{(2)}-m^{(2)}}$$

$$\tilde{\mathbf{O}}^{(2)} = \mathrm{diag}(e^{m^{(1)}-m^{(2)}})\tilde{\mathbf{O}}^{(1)} + e^{\mathbf{S}^{(2)}-m^{(2)}}\mathbf{V}^{(2)} = e^{s^{(1)}-m}\mathbf{V}^{(1)} + e^{s^{(2)}-m}\mathbf{V}^{(2)}$$

$$\mathbf{O}^{(2)} = \mathrm{diag}(\ell^{(2)})^{-1}\tilde{\mathbf{O}}^{(2)} = \mathbf{O}.$$

We describe the full FLASHATTENTION-2 forward pass in Algorithm 1.

---

**Algorithm 1** FLASHATTENTION-2 forward pass

---

**Require:** Matrices $\mathbf{Q},\mathbf{K},\mathbf{V} \in \mathbb{R}^{N \times d}$ in HBM, block sizes $B_c$, $B_r$.

1: Divide $\mathbf{Q}$ into $T_r = \left\lceil \frac{N}{B_r} \right\rceil$ blocks $\mathbf{Q}_1,...,\mathbf{Q}_{T_r}$ of size $B_r \times d$ each, and divide $\mathbf{K},\mathbf{V}$ in to $T_c = \left\lceil \frac{N}{B_c} \right\rceil$ blocks $\mathbf{K}_1,...,\mathbf{K}_{T_c}$ and $\mathbf{V}_1,...,\mathbf{V}_{T_c}$, of size $B_c \times d$ each.
2: Divide the output $\mathbf{O} \in \mathbb{R}^{N \times d}$ into $T_r$ blocks $\mathbf{O}_i,...,\mathbf{O}_{T_r}$ of size $B_r \times d$ each, and divide the logsumexp $L$ into $T_r$ blocks $L_i,...,L_{T_r}$ of size $B_r$ each.
3: **for** $1 \le i \le T_r$ **do**
4:     Load $\mathbf{Q}_i$ from HBM to on-chip SRAM.
5:     On chip, initialize $\mathbf{O}_i^{(0)} = (0)_{B_r \times d} \in \mathbb{R}^{B_r \times d}, \ell_i^{(0)} = (0)_{B_r} \in \mathbb{R}^{B_r}, m_i^{(0)} = (-\infty)_{B_r} \in \mathbb{R}^{B_r}$.
6:     **for** $1 \le j \le T_c$ **do**
7:         Load $\mathbf{K}_j,\mathbf{V}_j$ from HBM to on-chip SRAM.
8:         On chip, compute $\mathbf{S}_i^{(j)} = \mathbf{Q}_i\mathbf{K}_j^T \in \mathbb{R}^{B_r \times B_c}$.
9:         On chip, compute $m_i^{(j)} = \max(m_i^{(j-1)}, \mathrm{rowmax}(\mathbf{S}_i^{(j)})) \in \mathbb{R}^{B_r}$, $\tilde{\mathbf{P}}_i^{(j)} = \exp(\mathbf{S}_i^{(j)} - m_i^{(j)}) \in \mathbb{R}^{B_r \times B_c}$ (pointwise), $\ell_i^{(j)} = e^{m_i^{j-1} - m_i^{(j)}}\ell_i^{(j-1)} + \mathrm{rowsum}(\tilde{\mathbf{P}}_i^{(j)}) \in \mathbb{R}^{B_r}$.
10:         On chip, compute $\mathbf{O}_i^{(j)} = \mathrm{diag}(e^{m_i^{(j-1)} - m_i^{(j)}})\mathbf{O}_i^{(j-1)} + \tilde{\mathbf{P}}_i^{(j)}\mathbf{V}_j$.
11:     **end for**
12:     On chip, compute $\mathbf{O}_i = \mathrm{diag}(\ell_i^{(T_c)})^{-1}\mathbf{O}_i^{(T_c)}$.
13:     On chip, compute $L_i = m_i^{(T_c)} + \log(\ell_i^{(T_c)})$.
14:     Write $\mathbf{O}_i$ to HBM as the $i$-th block of $\mathbf{O}$.
15:     Write $L_i$ to HBM as the $i$-th block of $L$.
16: **end for**
17: Return the output $\mathbf{O}$ and the logsumexp $L$.

---

**Causal masking.**

One common use case of attention is in auto-regressive language modeling, where we need to apply a causal mask to the attention matrix $\mathbf{S}$ (i.e., any entry $\mathbf{S}_{ij}$ with $j > i$ is set to $-\infty$).

1. As FLASHATTENTION and FLASHATTENTION-2 already operate by blocks, for any blocks where all the column indices are more than the row indices (approximately half of the blocks for large sequence length), we can skip the computation of that block. This leads to around 1.7-1.8× speedup compared to attention without the causal mask.
2. We do not need to apply the causal mask for blocks whose row indices are guaranteed to be strictly less than the column indices. This means that for each row, we only need apply causal mask to 1 block (assuming square block).

**Correctness, runtime, and memory requirement.** As with FLASHATTENTION, Algorithm 1 returns the correct output $\mathbf{O} = \mathrm{softmax}(\mathbf{Q}\mathbf{K}^\top)\mathbf{V}$ (with no approximation), using $O(N^2 d)$ FLOPs and requires

$O(N)$ additional memory beyond inputs and output (to store the logsumexp $L$). The proof is almost the same as the proof of Dao et al. (2022, Theorem 1), so we omit it here.

### 3.1.2 BACKWARD PASS

The backward pass of FLASHATTENTION-2 is almost the same as that of FLASHATTENTION. We make a minor tweak to only use the row-wise logsumexp $L$ instead of both the row-wise max and row-wise sum of exponentials in the softmax. We include the backward pass description in Algorithm 2 for completeness.

**Multi-query attention and grouped-query attention.** Multi-query attention (MQA) (Shazeer, 2019) and grouped-query attention (GQA) (Ainslie et al., 2023) are variants of attention where multiple heads of query attend to the same head of key and value, in order to reduce the size of KV cache during inference. Instead of having to duplicate the key and value heads for the computation, we implicitly manipulate the indices into the head to perform the same computation. In the backward pass, we need to sum the gradients $\mathbf{dK}$ and $\mathbf{dV}$ across different heads that were implicitly duplicated.

### 3.2 PARALLELISM

The first version of FLASHATTENTION parallelizes over batch size and number of heads. We use 1 thread block to process one attention head, and there are overall batch size·number of heads thread blocks. Each thread block is scheduled to run on a streaming multiprocessor (SM), and there are 108 of these SMs on an A100 GPU for example. This scheduling is efficient when this number is large (say $\geq 80$), since we can effectively use almost all of the compute resources on the GPU.

In the case of long sequences (which usually means small batch sizes or small number of heads), to make better use of the multiprocessors on the GPU, we now additionally parallelize over the sequence length dimension. This results in significant speedup for this regime.

**Forward pass.** We see that the outer loop (over sequence length) is embarrassingly parallel, and we schedule them on different thread blocks that do not need to communicate with each other. We also parallelize over the batch dimension and number of heads dimension, as done in FLASHATTENTION. The increased parallelism over sequence length helps improve occupancy (fraction of GPU resources being used) when the batch size and number of heads are small, leading to speedup in this case.

**Backward pass.** Notice that the only shared computation between different column blocks is in update $\mathbf{dQ}$ in Algorithm 2, where we need to load $\mathbf{dQ}_i$ from HBM to SRAM, then on chip, update $\mathbf{dQ}_i \leftarrow \mathbf{dQ}_i + \mathbf{dS}_i^{(j)} \mathbf{K}_j$, and write back to HBM. We thus parallelize over the sequence length dimension as well, and schedule 1 thread block for each column block of the backward pass. We use atomic adds to communicate between different thread blocks to update $\mathbf{dQ}$.

We describe the parallelization scheme in Fig. 2.

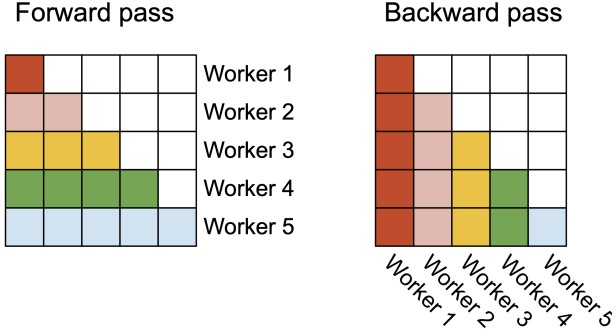

Figure 2: In the forward pass (left), we parallelize the workers (thread blocks) where each worker takes care of a block of rows of the attention matrix. In the backward pass (right), each worker takes care of a block of columns of the attention matrix.

**Decoding.** During LLM inference, most of the time is spent on iterative decoding, where one token is predicted at a time. The bottleneck for the attention operation during decoding is different from that during training or prefill (prompt processing), because the query length is very short (often query length is 1 since only the new extra token is attending to all the previous tokens, stored in the KV cache). As a

result, the bottleneck is no longer the read/write of intermediate matrices (the scores $\mathbf{QK}^\top$ and attention probabilities $\text{softmax}(\mathbf{QK}^\top)$). Instead, the bottleneck is to load the KV cache as quickly as possible.

To accommodate this setting, we split the KV cache loading among different thread blocks, to increase occupancy and saturate the HBM bandwidth. However, since the thread blocks cannot easily communicate with each other, we write intermediate results to HBM, then call a separate kernel to reduce the results and produce final output.

## 3.3 WORK PARTITIONING BETWEEN WARPS

As Section 3.2 describe how we schedule thread blocks, even within each thread block, we also have to decide how to partition the work between different warps. We typically use 4 or 8 warps per thread block, and the partitioning is described in Fig. 3.

**Forward pass.** For each block, FLASHATTENTION splits $\mathbf{K}$ and $\mathbf{V}$ across 4 warps while keeping $\mathbf{Q}$ accessible by all warps. Each warp multiplies to get a slice of $\mathbf{QK}^\top$, then they need to multiply with a slice of $\mathbf{V}$ and communicate to add up the result. This is referred to as the "split-K" scheme. However, this is inefficient since all warps need to write their intermediate results out to shared memory, synchronize, then add up the intermediate results. These shared memory reads/writes slow down the forward pass in FLASHATTENTION.

In FLASHATTENTION-2, we instead split $\mathbf{Q}$ across 4 warps while keeping $\mathbf{K}$ and $\mathbf{V}$ accessible by all warps. After each warp performs matrix multiply to get a slice of $\mathbf{QK}^\top$, they just need to multiply with their shared slice of $\mathbf{V}$ to get their corresponding slice of the output. There is no need for communication between warps. The reduction in shared memory reads/writes yields speedup (Section 4).

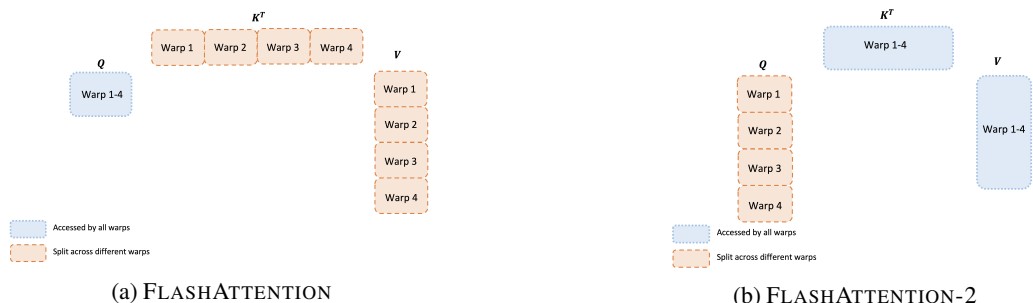

(a) FLASHATTENTION

(b) FLASHATTENTION-2

Figure 3: Work partitioning between different warps in the forward pass

**Backward pass.** Similarly for the backward pass, we choose to partition the warps to avoid the "split-K" scheme. However, it still requires some synchronization due to the more complicated dependency between all the different inputs and gradients $\mathbf{Q,K,V,O,dO,dQ,dK,dV}$. Nevertheless, avoiding "split-K" reduces shared memory reads/writes and again yields speedup (Section 4).

**Tuning block sizes** Increasing block sizes generally reduces shared memory loads/stores, but increases the number of registers required and the total amount of shared memory. Past a certain block size, register spilling causes significant slowdown, or the amount of shared memory required is larger than what the GPU has available, and the kernel cannot run at all. Typically we choose blocks of size $\{64, 128\} \times \{64, 128\}$, depending on the head dimension $d$ and the device shared memory size.

We manually tune for each head dimensions since there are essentially only 4 choices for block sizes, but this could benefit from auto-tuning to avoid this manual labor. We leave this to future work.

## 4 EMPIRICAL VALIDATION

We evaluate the impact of using FLASHATTENTION-2 to train Transformer models.

- **Benchmarking attention.** We measure the runtime of FLASHATTENTION-2 across different sequence lengths and compare it to a standard implementation in PyTorch, FLASHATTENTION, and FLASHATTENTION in Triton. We confirm that FLASHATTENTION-2 is 1.7-3.0× faster than FLASHATTENTION, 1.3-2.5× faster than FLASHATTENTION in Triton, and 3-10× faster than a standard attention implementation. FLASHATTENTION-2 reaches up to 230 TFLOPs/s, 73% of the theoretical maximum TFLOPs/s on A100 GPUs.

- **End-to-end training speed** When used end-to-end to train GPT-style models of size 1.3B and 2.7B on sequence lengths either 2k or 8k, FLASHATTENTION-2 yields up to 1.3× speedup compared

to FLASHATTENTION and 2.8× speedup compared to a baseline without FLASHATTENTION. FLASHATTENTION-2 reaches up to 225 TFLOPs/s (72% model FLOPs utilization) per A100 GPU.

## 4.1 BENCHMARKING ATTENTION FOR TRAINING

We measure the runtime of different attention methods on an A100 80GB SXM4 GPU for different settings (without / with causal mask, head dimension 64 or 128). We report the results in Fig. 4, Fig. 6 and Fig. 7, showing that FLASHATTENTION-2 is around 2× faster than FLASHATTENTION and FLASHATTENTION in xformers (the "cutlass" implementation). FLASHATTENTION-2 is around 1.3-1.5× faster than FLASHATTENTION in Triton in the forward pass and around 2× faster in the backward pass. Compared to a standard attention implementation in PyTorch, FLASHATTENTION-2 can be up to 10× faster.

Benchmark setting: we vary the sequence length from 512, 1k, ..., 16k, and set batch size so that the total number of tokens is 16k. We set hidden dimension to 2048, and head dimension to be either 64 or 128 (i.e., 32 heads or 16 heads). To calculate the FLOPs of the forward pass, we use:

$$4 \cdot \text{seqlen}^2 \cdot \text{head dimension} \cdot \text{number of heads}.$$

With causal mask, we divide this number by 2 to account for the fact that approximately only half of the entries are calculated. To get the FLOPs of the backward pass, we multiply the forward pass FLOPs by 2.5 (since there are 2 matmuls in the forward pass and 5 matmuls in the backward pass, due to recomputation).

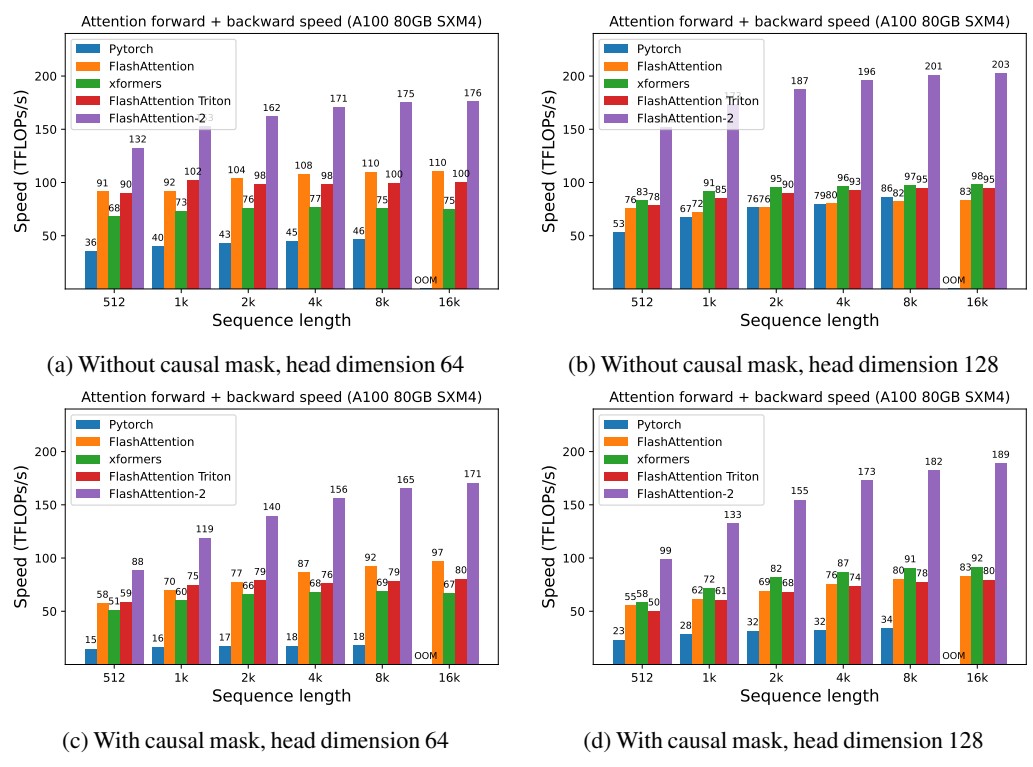

(a) Without causal mask, head dimension 64

(b) Without causal mask, head dimension 128

(c) With causal mask, head dimension 64

(d) With causal mask, head dimension 128

Figure 4: Attention forward + backward speed on A100 GPU

Just running the same implementation on H100 GPUs (using no special instructions to make use of new features such as TMA and 4th-gen Tensor Cores), we obtain up to 335 TFLOPs/s (Fig. 8). We expect that by using new instructions, we can obtain another 1.5x-2x speedup on H100 GPUs. We leave that to future work.

## 4.2 BENCHMARKING ATTENTION FOR INFERENCE

We benchmark the attention kernel during decoding for the case of multi-query attention, where the bottleneck is loading the KV cache. In Fig. 5, we see that the attention kernel from FLASHATTENTION-2 is up to 28× faster than a naive implementation in PyTorch, and up to 7× faster than an implementation

from FasterTransformer. This is thanks to better work partitioning where multiple thread blocks are loading the KV cache at the same time to saturate HBM bandwidth.

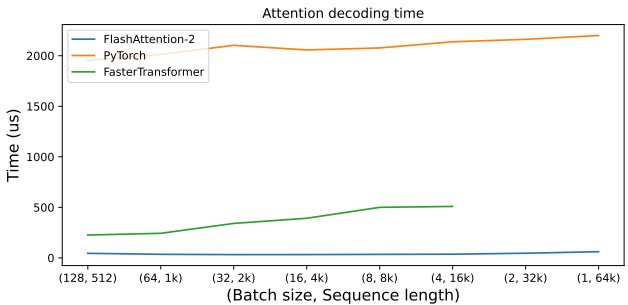

Figure 5: Attention decoding time on A100 80GB, with hidden dimension 2048 and multi-query attention. The attention kernel from FLASHATTENTION-2 is up to 7× faster than that of FasterTransformer and 28× faster than a naive implementation in PyTorch.

### 4.3 END-TO-END PERFORMANCE

We measure the training throughput of GPT-style models with either 1.3B or 2.7B parameters, on 8×A100 80GB SXM4. As shown in Table 1, FLASHATTENTION-2 yields 2.8× speedup compared to a baseline without FLASHATTENTION and 1.3× speedup compared to FLASHATTENTION, reaching up to 225 TFLOPs/s per A100 GPU.

Note that we calculate the FLOPs by the formula, following Megatron-LM (Shoeybi et al., 2019) (and many other papers and libraries):

$$6 \cdot \text{seqlen} \cdot \text{number of params} + 12 \cdot \text{number of layers} \cdot \text{hidden dim} \cdot \text{seqlen}^2.$$

The first term accounts for the FLOPs due to weight–input multiplication, and the second term accounts for the FLOPs due to attention. However, one can argue that the second term should be halved, as with causal mask we only need to compute approximately half the number of elements in attention. We choose to follow the formula from the literature (without dividing the attention FLOPs by 2) for consistency.

Table 1: Training speed (TFLOPs/s/GPU) of GPT-style models on 8×A100 GPUs. FLASHATTENTION-2 reaches up to 225 TFLOPs/s (72% model FLOPs utilization). We compare against a baseline running without FLASHATTENTION.

| Model | Without FLASHATTENTION | FLASHATTENTION | FLASHATTENTION-2 |
|---|---|---|---|
| GPT3-1.3B 2k context | 142 TFLOPs/s | 189 TFLOPs/s | 196 TFLOPs/s |
| GPT3-1.3B 8k context | 72 TFLOPS/s | 170 TFLOPs/s | 220 TFLOPs/s |
| GPT3-2.7B 2k context | 149 TFLOPs/s | 189 TFLOPs/s | 205 TFLOPs/s |
| GPT3-2.7B 8k context | 80 TFLOPs/s | 175 TFLOPs/s | 225 TFLOPs/s |

## 5 DISCUSSION AND FUTURE DIRECTIONS

FLASHATTENTION-2 is 2× faster than FLASHATTENTION, which means that we can train models with 16k longer context for the same price as previously training a 8k context model, for the same number of tokens. We are excited about how this can be used to understand long books and reports, high resolution images, audio and video. FLASHATTENTION-2 will also speed up training, finetuning, and inference of existing models.

In the near future, we plan to collaborate with researchers and engineers to make FlashAttention widely applicable in different kinds of devices (e.g., H100 GPUs, AMD GPUs), as well as new data types such as FP8. As an immediate next step, we plan to optimize FlashAttention-2 for H100 GPUs to use new hardware features (TMA, 4th-gen Tensor Cores, fp8). Combining the low-level optimizations in FlashAttention-2 with high-level algorithmic changes (e.g., local, dilated, block-sparse attention) could allow us to train AI models with much longer context. We are also excited to work with compiler researchers to make these optimization techniques easily programmable.

ACKNOWLEDGMENTS

We thank Phil Tillet and Daniel Haziza, who have implemented versions of FLASHATTENTION in Triton (Tillet et al., 2019) and the `xformers` library (Lefaudeux et al., 2022). FLASHATTENTION-2 was motivated by exchange of ideas between different ways that attention could be implemented. We are grateful to the Nvidia CUTLASS team (especially Vijay Thakkar, Cris Cecka, Haicheng Wu, and Andrew Kerr) for their CUTLASS library, in particular the CUTLASS 3.x release, which provides clean abstractions and powerful building blocks for the implementation of FLASHATTENTION-2. We thank Driss Guessous for integrating FLASHATTENTION to PyTorch. FLASHATTENTION-2 has benefited from helpful discussions with Phil Wang, Markus Rabe, James Bradbury, Young-Jun Ko, Julien Launay, Daniel Hesslow, Michaël Benesty, Horace He, Ashish Vaswani, and Erich Elsen. Thanks to Stanford CRFM and Stanford NLP for the compute support. We thank Dan Fu and Christopher Ré for their collaboration, constructive feedback, and constant encouragement on this line of work of designing hardware-efficient algorithms. We thank Albert Gu and Beidi Chen for their helpful suggestions on early drafts of this paper.

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

# A FLASHATTENTION-2 BACKWARD PASS

---

**Algorithm 2** FLASHATTENTION-2 Backward Pass

---

**Require:** Matrices $\mathbf{Q},\mathbf{K},\mathbf{V},\mathbf{O},\mathbf{dO} \in \mathbb{R}^{N \times d}$ in HBM, vector $L \in \mathbb{R}^N$ in HBM, block sizes $B_c$, $B_r$.

1: Divide $\mathbf{Q}$ into $T_r = \left\lceil \frac{N}{B_r} \right\rceil$ blocks $\mathbf{Q}_1,...,\mathbf{Q}_{T_r}$ of size $B_r \times d$ each, and divide $\mathbf{K},\mathbf{V}$ in to $T_c = \left\lceil \frac{N}{B_c} \right\rceil$ blocks $\mathbf{K}_1,...,\mathbf{K}_{T_c}$ and $\mathbf{V}_1,...,\mathbf{V}_{T_c}$, of size $B_c \times d$ each.

2: Divide $\mathbf{O}$ into $T_r$ blocks $\mathbf{O}_i,...,\mathbf{O}_{T_r}$ of size $B_r \times d$ each, divide $\mathbf{dO}$ into $T_r$ blocks $\mathbf{dO}_i,...,\mathbf{dO}_{T_r}$ of size $B_r \times d$ each, and divide $L$ into $T_r$ blocks $L_i,...,L_{T_r}$ of size $B_r$ each.

3: Initialize $\mathbf{dQ} = (0)_{N \times d}$ in HBM and divide it into $T_r$ blocks $\mathbf{dQ}_1,...,\mathbf{dQ}_{T_r}$ of size $B_r \times d$ each. Divide $\mathbf{dK},\mathbf{dV} \in \mathbb{R}^{N \times d}$ in to $T_c$ blocks $\mathbf{dK}_1,...,\mathbf{dK}_{T_c}$ and $\mathbf{dV}_1,...,\mathbf{dV}_{T_c}$, of size $B_c \times d$ each.

4: Compute $D = \mathrm{rowsum}(\mathbf{dO} \circ \mathbf{O}) \in \mathbb{R}^d$ (pointwise multiply), write $D$ to HBM and divide it into $T_r$ blocks $D_1,...,D_{T_r}$ of size $B_r$ each.

5: **for** $1 \leq j \leq T_c$ **do**

6:     Load $\mathbf{K}_j,\mathbf{V}_j$ from HBM to on-chip SRAM.

7:     Initialize $\mathbf{dK}_j = (0)_{B_c \times d}, \mathbf{dV}_j = (0)_{B_c \times d}$ on SRAM.

8:     **for** $1 \leq i \leq T_r$ **do**

9:         Load $\mathbf{Q}_i,\mathbf{O}_i,\mathbf{dO}_i,\mathbf{dQ}_i,L_i,D_i$ from HBM to on-chip SRAM.

10:         On chip, compute $\mathbf{S}_i^{(j)} = \mathbf{Q}_i \mathbf{K}_j^T \in \mathbb{R}^{B_r \times B_c}$.

11:         On chip, compute $\mathbf{P}_i^{(j)} = \exp(\mathbf{S}_{ij} - L_i) \in \mathbb{R}^{B_r \times B_c}$.

12:         On chip, compute $\mathbf{dV}_j \leftarrow \mathbf{dV}_j + (\mathbf{P}_i^{(j)})^\top \mathbf{dO}_i \in \mathbb{R}^{B_c \times d}$.

13:         On chip, compute $\mathbf{dP}_i^{(j)} = \mathbf{dO}_i \mathbf{V}_j^\top \in \mathbb{R}^{B_r \times B_c}$.

14:         On chip, compute $\mathbf{dS}_i^{(j)} = \mathbf{P}_i^{(j)} \circ (\mathbf{dP}_i^{(j)} - D_i) \in \mathbb{R}^{B_r \times B_c}$.

15:         Load $\mathbf{dQ}_i$ from HBM to SRAM, then on chip, update $\mathbf{dQ}_i \leftarrow \mathbf{dQ}_i + \mathbf{dS}_i^{(j)} \mathbf{K}_j \in \mathbb{R}^{B_r \times d}$, and write back to HBM.

16:         On chip, compute $\mathbf{dK}_j \leftarrow \mathbf{dK}_j + {\mathbf{dS}_i^{(j)}}^\top \mathbf{Q}_i \in \mathbb{R}^{B_c \times d}$.

17:     **end for**

18:     Write $\mathbf{dK}_j,\mathbf{dV}_j$ to HBM.

19: **end for**

20: Return $\mathbf{dQ},\mathbf{dK},\mathbf{dV}$.

---

# B BENCHMARKING ATTENTION ON A100 AND H100

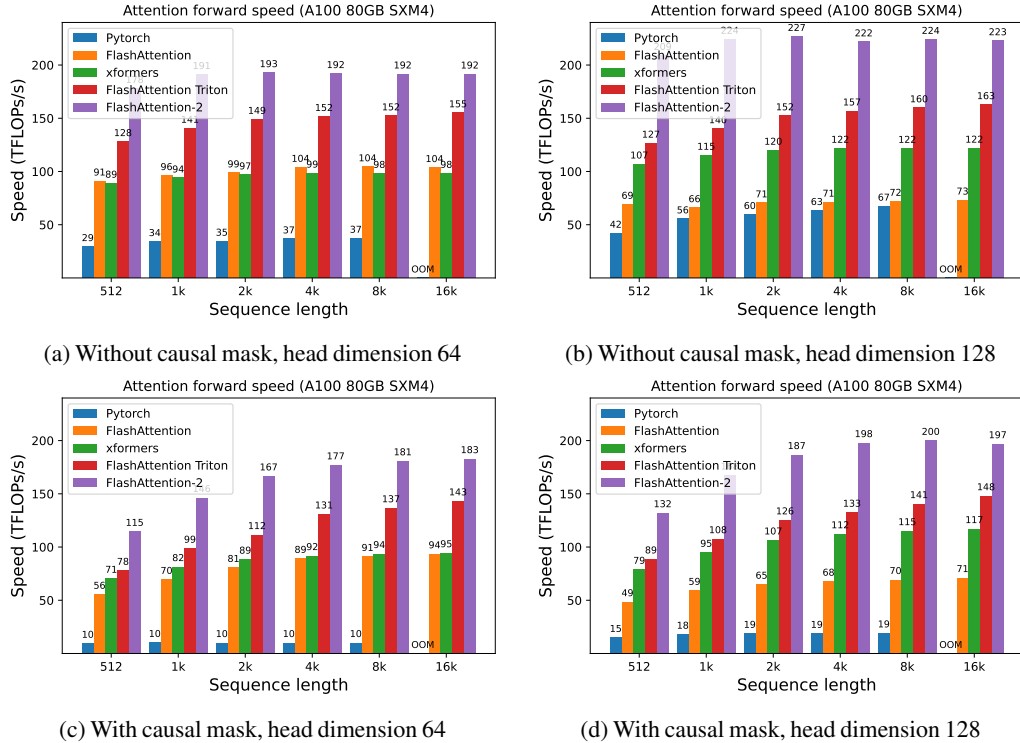

(a) Without causal mask, head dimension 64

(b) Without causal mask, head dimension 128

(c) With causal mask, head dimension 64

(d) With causal mask, head dimension 128

Figure 6: Attention forward speed on A100 GPU

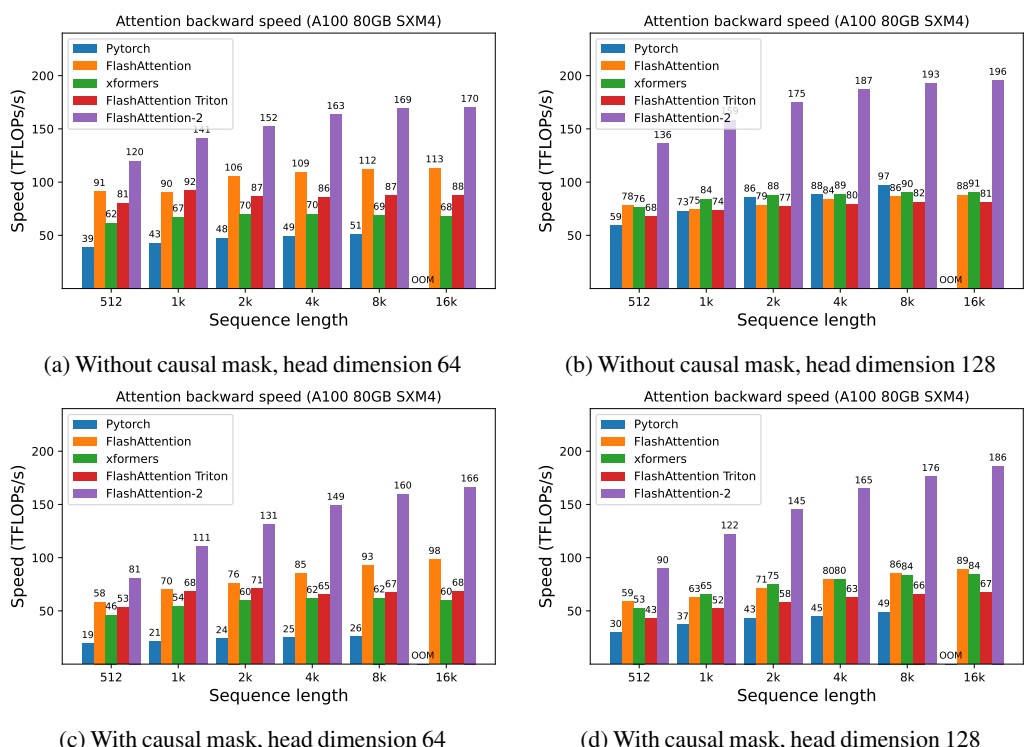

(a) Without causal mask, head dimension 64

(b) Without causal mask, head dimension 128

(c) With causal mask, head dimension 64

(d) With causal mask, head dimension 128

Figure 7: Attention backward speed on A100 GPU

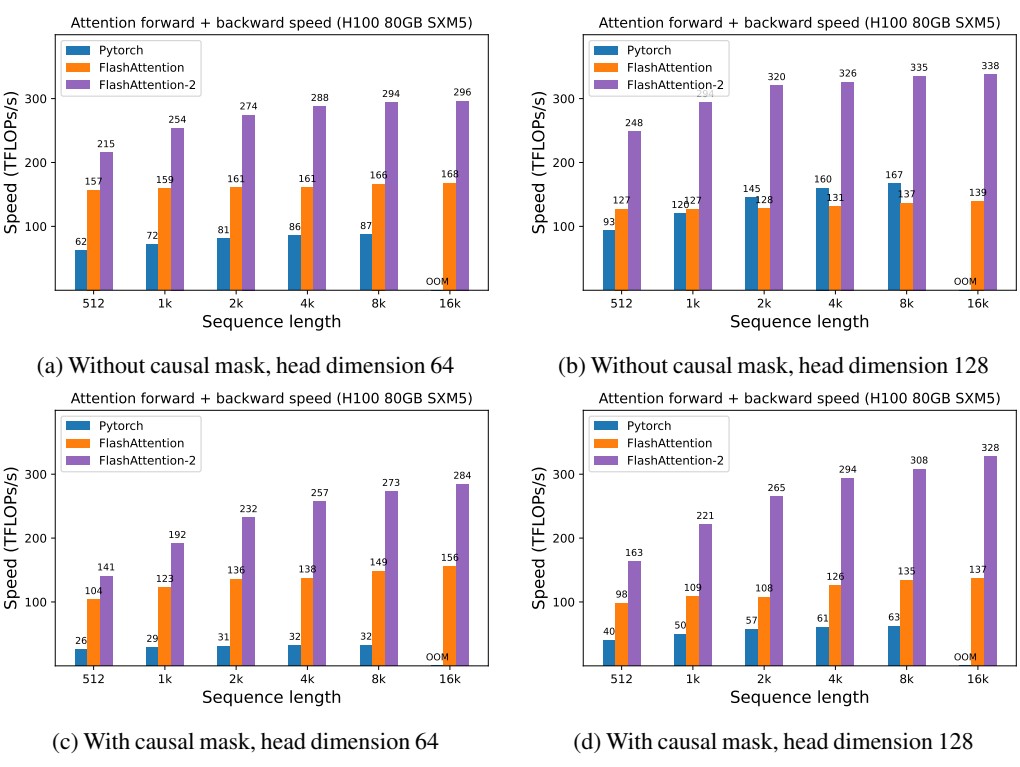

(a) Without causal mask, head dimension 64

(b) Without causal mask, head dimension 128

(c) With causal mask, head dimension 64

(d) With causal mask, head dimension 128

Figure 8: Attention forward + backward speed on H100 GPU

