# OpenReview forum: "FlashAttention-2: Faster Attention with Better Parallelism and Work Partitioning"
_ICLR.cc/2024/Conference — ICLR 2024 poster_

### Official Review · Reviewer_Jn9J · 2023-11-04

**Soundness:** 2 fair
**Presentation:** 2 fair
**Contribution:** 2 fair
**Rating:** 5
**Confidence:** 3

**Summary:**

This paper describes FlashAttention-2 which improves upon FlashAttention by introducing "tweaks" to improve performance on GPUs.   The paper claims the tweaks improve performance by increasing occupancy and use of matrix-multiply hardware (tensor cores).  The paper reports a bit under $1.3\times$ wall clock speedup versus FlashAttention.

**Strengths:**

Improving training speed of LLMs is of great interest to many.

**Weaknesses:**

Could do a better good job explaining how a given "tweak" helps achieve a given improvement (occupancy, use of tensor cores).

**Questions:**

Regarding the equation at the top of Page 5, I am unclear "$\mbox{diag}(l^{(1})^{-1}$" is to the power -1.  Comparing to the prior equation seems like exponent of -1 should be 1.

I think it would help some readers (like me) understand the contribution a bit better if the paper briefly summarized the key changes in the six (unnumbered) equations on Page 5 that are described as the "online softmax trick" versus the six on Page 3.

How do the "tweaks" in Section 4.1.1 help reduce non-matrixmul FLOPs?  I know a fair amount about tensor cores, but it wasn't obvious to me.

The paper claims occupancy is increased on Page 6 but it was unclear: (i) what definition of occupancy is being used (GPU resources could mean many things and occupancy often just refers to number of warps that can concurrently run versus max number supported by hardware ); and (ii) whether any measurement has been made to confirm the claimed improvement (e.g., using NVIDIA Parallel Nsight or similar approaches for collecting performance counters).

Much of Algorithm 1 seems similar to the original FlashAttention.  It may help summarizing which lines are different.  It would also help the reader if there was a summary of which lines lead to the reduction in non-matrixmul FLOPs and improved occupancy.

"Only at the every end of the" - typo.

For the backward pass (Section 3.1.2): It was unclear what the relevance of the paragraph on MQA and GQA is to the changes in FlashAttention-2 versus FlashAttention.

In Figure 2, does an uncolored square mean no computation?  Does the backward pass for a given worker start right away or do workers need to synchronize between forward and backward pass?  Do you not need to compute the combined result for the forward pass before you can start the backward pass?    If you do need to wait, then how can one achieve greater than 50% use of peak performance if roughly half the compute cycles are spent waiting for the longest running forward/backward pass thread block to complete?   If you don't need to wait, why not?

I'm not sure how to relate Figure 3 to Algorithm 1 (i.e., which lines it is meant to illustrate).  From the two paragraphs above Figure 3 I get it there are two potential sources of reduced execution time: fewer shared memory accesses and fewer synchronizations (__syncthreads, I assume).  Unclear which of those matters most and why given that shared memory accesses proceed about as fast as register file accesses and synchronization with a thread block is low overhead.

Why is FlashAttention (version 1) missing in Figure 5?

As someone who knows GPUs well, I would have liked to see more performance counter data to backup the claims of the sources of performance improvements.   I understand space is limited in the main text, but in checking the supplemental material, while it is great to see all the code, there appeared to be no PDF providing additional data or details.  Including one might have helped.

**Details Of Ethics Concerns:**

Hard to say this submission conforms to double blind review standards given the title of the paper strongly suggests author overlap with the prior FlashAttention work.   Given the nature of unconscious bias, I cannot say whether this may have influenced me and if so how (which is the whole purpose of using double blind review).

---

> ### Author Response · Authors · 2023-11-23
> **Reponse**
>
> Thank you for the constructive feedback to help improve our work!
>
> Q: How does the tweaks reduce non-matmul FLOPs.
>
> A: The amount of matmul FLOPs stay the same (e.g. Q @ K^T and attn_prob @ V). However, FlashAttention incurs more vector flops to rescale the output (equation at bottom of page 4): extra divisions by l^2 on both terms. FlashAttention2 gets rid of this division (top equation at top of page 5).
>
> In addition, masking (non-matmul operation) is only done to 1 block and not every blocks, again reducing non-matmul FLOPs so that most of the time is spent on matmul FLOPs accelerated by tensor cores.
>
> These changes generally reduce register pressure as an added benefit.
>
> Q: Figure 2, does an uncolored square mean no computation?
>
> A: Yes, we skip those blocks since they don't contribute to the outputs (i.e. masked out) any way.
>
> Q: backward pass synchronization with forward pass.
>
> A: The backward pass is done separately from the forward pass, so there's no overlap between them. As an example, in the model forward pass, the attention and MLP forward pass are performed, then we calculate a loss function, which then gives a gradient that gets propagated to the MLP backward pass and then attention backward pass.
>
> Q: Figure 3.
>
> A: Thanks for this clarification, we will add this to the paper. Figure 3 corresponds to line 8 and 10 in Algorithm 1: matmul Q K^T and matmul P V. Figure 3 illustrates how to map these matrix multiplications to 4 warps in the same threadblock.
>
> Q: FlashAttention in Figure 5.
>
> A: For decoding with long sequences, FlashAttention is actually slower than Pytorch due to lack of parallelism. As a result FlashAttention was only used for prompt processing (prefill) and not decoding (KV cache). FlashAttention-2 now speeds up both prompt processing (prefill) and decoding.
>
> We will add this note to the paper.
>
> Q: performance counter data.
>
> A: Thanks for this great suggestion. We will add this to the paper to help clarify the effectiveness of our approaches.

---

### Official Review · Reviewer_hPzR · 2023-11-05

**Soundness:** 2 fair
**Presentation:** 3 good
**Contribution:** 3 good
**Rating:** 8
**Confidence:** 5

**Summary:**

The authors present a new algorithm, FlashAttention-2, which builds in FlashAttention to improve the efficiency of the attention algorithm when executed on GPUs. The authors focus, in particular, on maximizing the amount of time spent in "matrix-multiply" FLOPs, that is, computation that is using matrix multiplication units, which are better lent to GPU hardware given particular division of work amongst warps and loose requirements around shared memory accesses.

These optimizations include:
1. Deferring scaling of values in the online softmax computation to further reduce HBM utilization
2. Save a logsumexp for online softmax backwards rather than max and sums to reduce the memory usage.
3. The bulk of the changes present in FlashAttention-2 are related to scheduling. Parallelization over the sequence dimension results in better warp occupancy in cases where there are few attention heads or a low batch size. Further, the authors change how the backward pass shares computation in the query derivative update, which also reduce HBM utilization. Splitting the KV cache amongst thread blocks also helps to saturate memory bandwidth. Better partitioning amongst warps, overall, drives better utilization.

**Strengths:**

- The proposed approach shows promising improvements in a performance-critical parts of end-to-end transformer computation.
- The approach is symmetrically applicable for both training and inference with broad applications in both research and production settings.
- It is important that engineering contributions such as FlashAttention-2 (which I will refer to FA-2) are part of conference literature, and the attention to detail therein is what drives the solid impact of this work rather than a special "algorithmic contribution" in a classical sense.
- The devil is in the details: careful analysis of scheduling and an intuitive approach to laying out computation drives the approach's strong results.
- The work builds on an already-strong baseline, FlashAttention.
- The paper is well-written and organized and clearly lays out the authors' contributions; in particular, the paper is quite accessible to those without a low-level background in machine learning computation or GPU programming.

**Weaknesses:**

- The baselines benchmarked in the paper can be stronger.
  - For latency benchmarks in Figure 5, the only baselines are a FasterTransformer and PyTorch. The authors do not consider compilers
  - Do the PyTorch benchmarks in Figure 5 use CUDA Graphs? PyTorch has significant framework overhead, and CUDA Graphs can give an order of magnitude speedup for some workloads, especially latency-sensitive ones.
- The above applies more generally to the other evaluation in Section 4; FlashAttention-2 is compared to PyTorch, then implementations with Triton and Cutlass, but not with any other frameworks capable of code generation. For example: while XLA may not be memory-bandwidth-aware by default, it can still generate kernels with fused operators that significantly reduce total memory I/O.
- While not needing to resort to approximations is a significant advantage of FlashAttention-2, this could be highlighted much more in the manuscript. Section 1 discusses many alternative attention approximations -- even speculation about why these aren't used (i.e. they are riskier when researchers have limited resources and don't adapt as easily) would strengthen the exactness boon of the authors' approach.

Several improvements to writing might improve the paper:
- The constant and equation in general in Sections 4.1 (i.e. 4) is not adequately explained (why is the sequence length squared? why 4?). Clarifying these might help new readers.
- The usage of "major problem" in the first sentence of the abstract is unclear -- it's clear that scaling sequence lengths is difficult; are the authors suggesting the problem is difficult, significant, or both?
- Section 1: "However, context length increases" <-- is missing "**as** context length increases"
- Text in all of the provided diagrams can be made clearer, and the diagrams can be rendered more clearly. It is difficult to read them as is.
- Section 3.2 might more clearly explain "prefill" and "KV cache" to readers. While somewhat ubiquitously understood amongst people doing performance engineering for large-scale transformers, some clarification would help the paper flow and increase its accessibility.
- Figure 2 can be clearer with respect to rows and columns -- this is the attention matrix -- what is its size/can the axes be labeled?

**Questions:**

- Why are the gaps between FA/FA-2 different with Cutlass versus in Triton, if the authors were to speculate? Further, the comparison in section 4.1 can be clarified -- is the assertion that the performance of the vanilla CUDA implementation of FlashAttention and the Cutlass FlashAttention implementation in xformers are congruent?
- The authors might also consider mentioning in the manuscript what they think trends in changing GPU hardware will mean for FA-2's general direction. Given that HBM bandwidth is not improving as quickly as SM arithmetic latency and that the amount of vram available on GPUs is not increasing, how will the approaches used in FA-2 change in relevance over time?
- Do you think a compiler could realistically generate FA-2? What sort of cost models might be required?
- How does FA-2 function when there is no explicit sequence length dimensions? In many training setups, tokens are padded without sequence boundaries, and models learn end-of-sequence tokens implicitly. What is the default behavior in this regime?

---

> ### Author Response · Authors · 2023-11-23
> **Response**
>
> Thank you for the detailed and constructed feedback.
>
> Q: Figure 5 baselines.
>
> A: Here we measure just the attention decoding operation, not the end-to-end latency. We agree that CUDA graph helps end-to-end generation speed, especially for small models where the CPU overhead can be large.
> We have benchmarked the kernel generated by torch.compile which is faster than Pytorch eager but still not as fast as the hand-written kernel from FasterTransformer. FA2 decoding kernel is faster still.
>
> Q: Comparison to compilers.
>
> A: Compilers can generally perform fusion. However, optimizations that require mathematical rewriting of the same expression (while maintaining numerical stability) are generally harder for compilers. Jax uses the powerful XLA compiler, but currently the best implementation of attention on TPUs is a version of FA2 implemented in Pallas, where the programmer still specifies some details (which elements should be in HBM vs SRAM) leaving the Pallas compiler to generate code to invoke the systolic array for matmuls or vector units for other computation.
>
> Q: gap between different implementations, e.g. xformers, Cutlass, Triton.
>
> xformers uses cutlass to implement a similar algorithm to FA, while Triton generates ptx directly. The differences in speed are due to low-level implementation: e.g. the Triton compiler only deals with power-of-2 block sizes, while with cutlass one can use non-power-of-2 block sizes. The triton compiler will use async memory copy automatically, while with Cutlass one has to spend more effort to use these hardware features.
>
> Q: Change in hardware trend.
>
> This is a great question. HBM bandwidth not increasing as fast as matmul FLOPs means that techniques such as FA/FA2 will be even more relevant in the future. New hardware features (asynchronous computation, dataflow architecture) might require new ideas to efficiently use the hardware, and is an exciting future area.
>
> Q: FA2 with variable sequence lengths.
>
> FA2 does support variable sequence lengths. Sequences are concatenated with no padding token, with an array indicating the lengths of each sequences. This is convenient for settings where sequences can have different lengths, and avoids wasting computation on padded tokens.

---

### Official Review · Reviewer_RLZf · 2023-11-07

**Soundness:** 4 excellent
**Presentation:** 3 good
**Contribution:** 4 excellent
**Rating:** 10
**Confidence:** 4

**Summary:**

This paper presents improvements to FlashAttention (Dao et al., 2022), a established method for efficiently computing attention through fused ops. These improvements are designed specifically for better parallelism and work partitioning in GPUs, resulting in the development of FlashAttention v2. Performance benchmarks have been conducted for both training and inference phases. Additionally, the authors provide comprehensive results from training end-to-end GPT-style models with 1.3 billion and 2.7 billion parameters and 2k and 8k context sizes.

**Strengths:**

There are many things to like in this paper, such as:

1. The paper is well-written
2. The improvements are well-explained and justified
3. The paper covers both training and inference time optimization
4. The results are encouraging

In summary, I expect widespread adoption of FlashAttention v2 within the community. Furthermore, the methods proposed and utilized in this paper could inspire the creation of more efficient components in machine learning.

**Weaknesses:**

FlashAttention v2 has a notable limitation: it relies on recent, specialized GPU architectures like the A100 (and H100). Additionally, the requirement for custom CUDA kernels adds a layer of complexity.

A small critique is that Figure 3 could benefit from a more descriptive caption.

**Questions:**

Which GPU architectures currently support FlashAttention v2?
What are the minimum requirements for its use?
What modifications are necessary to adapt FlashAttention v2 for use with relative positional encoding methods (e.g., RoPE and ALiBi)?
Does FlashAttention v2 offer compatibility with sparse block masks (as in v1)?

---

> ### Author Response · Authors · 2023-11-23
> **Response**
>
> Thank you for the very encouraging feedback!
>
> Q: Which GPUs are supported?
>
> A: FA2 supports Ampere (e.g, RTX 3090, A100), Ada (e.g. RTX 4090), and Hopper GPUs (H100). FA2 has also been independently implemented on TPUs, AMD GPUs, and Intel CPUs. We included more details in the common response.
>
> Q: Relative position encoding support?
>
> RoPE is not technically part of the core attention operation (applied to Q and K before the inner attention). We do provide an optimized (fused) implementation of RoPE as part of the FA2 library. In addition, FA2 now supports ALiBi.

---

### Official Review · Reviewer_AK2B · 2023-11-10

**Soundness:** 4 excellent
**Presentation:** 3 good
**Contribution:** 3 good
**Rating:** 6
**Confidence:** 4

**Summary:**

Scaling Transformers for longer sequences holds the promise of enhancing language modeling and understanding complex inputs, but is hindered by the attention layer's quadratic scaling in memory and runtime. FlashAttention has mitigated this by bringing linear memory usage and considerable runtime speedup, yet it still lags behind the efficiency of optimized matrix multiplication operations. To address this, FlashAttention-2 is introduced with improved work partitioning, yielding a significant speedup and reaching closer to the efficiency of matrix multiply (GEMM) operations. Empirical validation shows that FlashAttention-2 significantly increases the training speed of GPT-style models on both A100 and H100 GPUs.

**Strengths:**

* The proposed platform-specific optimizations are clever and sound.
* The resulting software artifacts are useful and have has the potential to benefit both researchers and practitioners.

**Weaknesses:**

* The work is mostly engineering-focused, with several "tweaks" made to FlashAttention.
* The performance gains are relatively marginal, especially when compared to those of the original FlashAttention over the baseline.
* The absence of an ablation study makes it difficult to pinpoint the exact sources of efficiency.

**Questions:**

Thank you for submitting to ICLR 2024. FlashAttention-2 is a very useful artifact that has the potential to benefit both researchers and practitioners, and the proposed optimization techniques appear sound.

Here are my questions I would like the authors to answer:
* Perhaps the most significant omission in this paper is the lack of an ablation study. This makes it challenging to discern the contributions of individual optimizations. Among the proposed optimizations, which one has the highest impact?
* In Section 3.1, is the technique of skipping blocks for "causal masking" also applied to FlashAttention? As the authors mention, this technique can be applied to both FlashAttention and FlashAttention-2, and I am curious about how the application of this technique would affect the performance gap between FlashAttention and FlashAttention-2 if it had not been applied to FlashAttention.
* In Section 3.3, what is the performance impact of "tuning block sizes"? Was the same level of parameter tuning effort applied to FlashAttention? My question concerns the extent to which the performance gains over FlashAttention can be attributed to algorithmic improvements versus additional tuning effort, with the latter being less significant.

---

> ### Author Response · Authors · 2023-11-23
> **Response**
>
> Thank you for the helpful suggestions on the ablation study to improve the work. We have included such a study in the common response.
>
> Q: The performance gains are relatively marginal, especially when compared to those of the original FlashAttention over the baseline.
>
> A: Attention is now one of the most optimized components as one of the core layers of Transformers training on massive clusters. FlashAttention-2 speeding up attention by up to 2x can directly benefit these large training runs (costing millions of dollars). It can also enable new capabilities such as long context models (for books and code, high-res images, audio signal). It also make it easier to deploy models locally on desktop GPUs or enable research into these models due to faster iteration speed.
>
> Q: Is skipping blocks for "causal masking" also applied to FlashAttention?
>
> A: Yes, in our benchmarks, we also apply the same technique to FlashAttention. The difference in speed is due to our 3 contributions.
>
> Q: What is the performance impact of "tuning block sizes"?
>
> This was done at the very end to squeeze out a little more performance, on the order of 5%. It was not done with FlashAttention since its implementation was much more rigid and the block sizes were hard-coded. We do not expect block size tuning to play a major role here.

---

### Author Response · Authors · 2023-11-23
**Common response**

We thank the reviewers for insightful comments and constructive feedback. We are happy that reviews were positive, and that reviewers thought that our work addresses an important problem  and results in useful software artifacts benefiting researchers and practitioners, and that our paper was clear and well-written.

We first report some updates, address a few common questions, and then respond to specific questions from the reviewers.

We are happy to report several updates to FlashAttention-2 (FA2) since submission. We are happy to see that FA2 has already begun making an impact in a short time:
* **Getting FlashAttention-2 into people’s hands**: we have been working with the Pytorch and Huggingface developers to integrate FA2 into these widely-used libraries. FA2 is available through torch.nn.Transformer and torch.nn.functional.scaled_dot_product_attention (available now on torch-nightly, slated for PyTorch 2.2 release), as well as Huggingface’s implementation of many popular models (e.g. Llama) in the `transformers` library. FA2 is also integrated into many inference libraries (Nvidia’s TensorRT-LLM, Huggingface’s Text Generation Inference), as well as many finetuning libraries (fastchat, axolotl). This has benefitted a large audience of researchers and practitioners, as well as AI enthusiasts and developers building applications on top of large language models and diffusion models.

* **Implementations on other hardwares**: Though our work presented in this paper focuses on Nvidia GPUs, the general ideas can apply to other hardware accelerators. FA2 has been independently implemented in Jax (through Pallas, a high-level language embedded in Python) that can run on both Nvidia GPUs and TPUs. Thanks to the effort of the OpenAI Triton team and developers from AMD and Intel, FA2 is also independently implemented in the Triton language with backends that run on AMD GPUs (e.g. MI250 and MI300) and Intel CPUs and GPUs.

* **Latest MLPerf training benchmark**: MLPerf is an industry-wide benchmark on the latest hardware and best software implementations of ML models. In MLPerf training 3.1 (Oct-Nov 2023), FA2 was a component of the fastest GPT3 (175B) implementation on H100 GPUs and on TPU v5e, as well as a component to achieve the fastest training of the Stable Diffusion v2 model on H100 GPUs. We are happy to see that our conceptually simple approach is making progress on a well-studied problem, benefiting several important applications.

* **Collaborations on open source models**: We have collaborated with several organizations to add new features to FA2 and use it to train open source models. As an example, FA2 with local (i.e. sliding-window) attention was used to train the Mistral-7B model, arguably one of the strongest open source LLMs currently. Mistral-7B enabled AI researchers and developers to study and deploy much smaller open models with comparable capabilities to larger-scale or proprietary models.

We now address some common questions about our contributions.

**Ablation: contribution of different techniques**
We ablate the effectiveness of our three techniques: (1) fewer non-matmul FLOPs (2) threadblock parallelism (3) warp partitioning to reduce communication. Threadblock parallelism is most important in the case of small batch size and number of heads (common in training large models distributed across many accelerators), since FlashAttention suffers from low occupancy there (not using all the streaming multiprocessors on the GPUs due to not enough parallel work). In other cases, better warp partitioning plays a major role in speeding up the computation, since one of the bottlenecks of FlashAttention is communication between different warps through shared memory. Fewer non-matmul FLOPs are always helpful.

We measure speed for the attention forward pass, with causal mask, head dimension 128, sequence length 2048, on an A100 80GB SXM4 GPU. We consider two setting:

(A) large batch size x number of heads: we set batch size = 8 and number of heads = 16

| Method                                     | Speed (TFLOPs/s) |
|-------------------------------------------- | ------------------|
| FlashAttention                             | 65               |
| + (1) Fewer non-matmul FLOPs               | 91 (1.4x)        |
| + (1) + (2) Threadlbock paralelism         | 110 (1.2x)       |
| + (1) + (2) + (3) Better warp partitioning | 187 (1.7x)       |

(B) small batch size x number of heads: we set batch size = 1 and number of heads = 16

| Method                                     | Speed (TFLOPs/s) |
|-------------------------------------------- | ------------------|
| FlashAttention                             | 23               |
| + (1) Fewer non-matmul FLOPs               | 32 (1.4x)        |
| + (1) + (2) Threadlbock paralelism         | 103 (3.2x)       |
| + (1) + (2) + (3) Better warp partitioning | 143 (1.4x)       |

---

### Public Comment · ~Zhenan_Wu1 · 2024-06-19

In page 3 it is stated dV=P^T dO, but from page 2 we have O=PV, so doesn't this imply dV=P^T PdV, and that P^T P=I?

---

> ### Public Comment · ~Tri_Dao1 · 2024-06-19
>
> No, dO is short for $\frac{d \mathrm{loss}}{d O}$, i.e. gradient wrt O, not gradient of O wrt some other variable.

---

> > ### Public Comment · ~Zhenan_Wu1 · 2024-06-19
> >
> > thanks for the clarification

---

### Meta-Review · Area_Chair_wq4h · 2023-12-14

**Metareview:**

This paper proposes an improved version of FlashAttention which is more efficient than the original one. Most reviewers agree that this is a strong paper, emphasizing clarity, engineering work, empirical validation, and demonstrated practical usefulness. Even though the novelty beyond the original FlashAttention is somewhat limited, the proposed method is already being adopted by the community and seems to be having real impact. I recommend acceptance.

**Justification For Why Not Higher Score:**

Even though the proposed FlashAttention2 is already being impactful, there is not enough scientific novelty in this paper to justify (in my opinion) being highlighted as a spotlight paper.

**Justification For Why Not Lower Score:**

Already stated in the review.

---

### Decision · Program_Chairs · 2024-01-16

Accept (poster)